# Typology and Unified Model of the Sharing Economy in Open Innovation Dynamics

**Yeji Kim and Minhwa Lee \***

KCERN, Seoul 06301, Korea; yeji8793@gmail.com
* Correspondence: minhwalee@kaist.ac.kr

**Abstract:** The sharing economy is emerging as one of the hottest issues of the Fourth Industrial Revolution. The ownership paradigm was dominant during the First and Second Industrial Revolutions, so the formation of the sharing economy was almost non-existent, but it has grown to 5% of the global GDP during the Third Industrial Revolution as the sharing paradigm became prominent. It is expected that the scale of the sharing economy will reach up to 50% of the global economy by 2025 as the online to offline convergence (O2O) phenomenon (GE, 2012). The sharing economy is generally considered complex, diverse, and simply chaotic territory due to its various meanings or types despite its importance. In short, there is a great need to do more research to develop a unified model of the sharing economy. Our study defines sharing economy as "an activity where economic agents share economic objects together to create values". The KCERN Sharing Economy Cube Model presented in the study is a unified model where the subjects of sharing—supply, market platform, demand, etc.—share the objects of sharing—information, materials, relations, etc.—in order to engage in economic activities, both for profit and nonprofit, to create values. The model reflects all these activities and encompasses all the other definitions of the sharing economy. This study aims to systematically draw a roadmap for the national sharing economy in the ongoing Fourth Industrial Revolution era based on the integrative sharing platform economy model.

**Keywords:** sharing economy; sharing economy cube model; Fourth Industrial Revolution; sharing economy national strategy; platform; open innovation

## 1. Introduction

The sharing economy became clearly visible with the advent of Uber and Airbnb, and now the world's top 10 companies with the largest total sale value and top 20 unicorn companies are on the sharing platform with 70% of their total market value. An expansion of sharing economy is related to an evolution of the Industrial Revolution since the evolution has drawn the fusion of physical space dominated by ownership and virtual space oriented towards sharing [1].

The sharing economy is not a new concept. Locale-based sharing activities or cooperatives, which were based on offline platforms, were active during the First and Second Industrial Revolutions. These activities could also be considered as an earlier form of sharing economy, but their scales or effects were weak. It was the development of the wired and wireless internet during the Third Industrial Revolution that expanded the extent of the sharing economy based on the online platform. Therefore, the full-scale sharing economy first appeared on the scene.

The Fourth Industrial Revolution that combines offline and online led to the explosive growth of the sharing economy all over the world; it is expected that the value of the companies in the sharing economy, which was only 2.6 billion dollars now, will increase up to 335 billion dollars in 2025 through industries such as crowd funding, P2P lodging, car-sharing services, etc. [2]. The size of the sharing economy is expected to be comparable to the existing rental market [3]. From traditional industry

where industrial internet accounted for about 46% of global GDP, it is estimated that sharing economy industries will take up 50% of the global economy [4].

As the sharing economy experiences a rapid rise, new sharing economy business models are appearing in diverse areas. However, there is no commonly agreed-upon definition of sharing economy. Therefore, the ultimate purpose of this study is to answer the following five research questions as below.

How can we define sharing economy?

Why do we need the unified definition of sharing economy and what is the unified definition?

How did the Industrial Revolution impact on sharing economy?

Why do we need to relate the sharing economy and the Industrial Revolution?

How can we respond to boost the sharing economy and make national policies?

This study used methods of qualitative systematic review to answer these research questions because analyzing previous study can help to understand unified thinking. Furthermore, this study introduces KCERN Sharing Economy Cube Model in order to form the fundamental definition of share economy by reviewing previous researches. The contribution of this study is to respond concretely to such big trends of the sharing economy and to let public organizations develop strategic policies.

The present study is organized as follows. Section 2 shows research methods, and Section 3 summarizes previous researches on the definition and concept of sharing economy and analyzes their limitations. In addition, KCERN Sharing Economy Cube Model and its contributions are suggested, and accordingly national strategies to facilitate sharing economy are drawn [5]. Finally, a summary and conclusion of the study are explained.

## 2. Methods

This study gives out five questions as hypotheses and reviews previous literature to analyze and organize typology of the sharing economy. Based on review, this study develops a unified model and insists on the practical advantages of it. Accordingly, national policies and strategies will be drawn at last. Process of this study is seen in Figure 1.

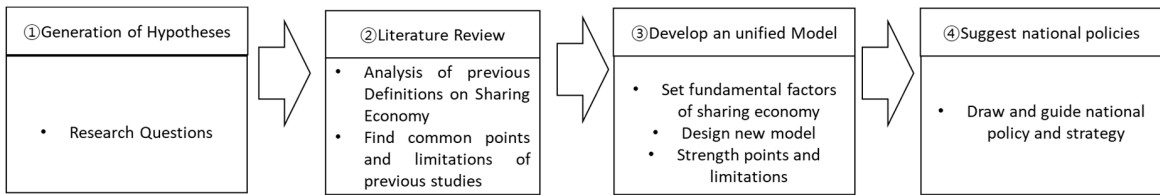

**Figure 1.** Methods and process.

For systematic analysis, this study conducted qualitative review using PRISMA statement and flowchart at most as seen in Figure 2 [6]. PRISMA is preferred reporting items for systematic reviews for meta-analysis. At first, this study searched all relevant studies and decided eligibility criteria in order to select studies while excluding some reviews for reasons such as duplication. Thirdly, results of individual studies and synthesis of results were drawn. Then, limitations of previous studies and conclusions can be conveyed.

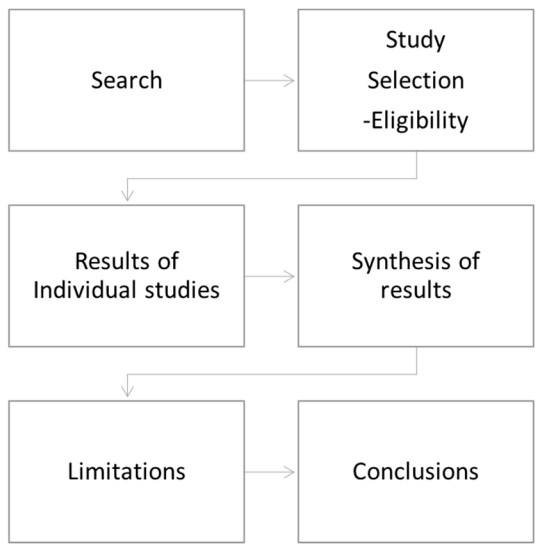

**Figure 2.** Systematic analysis flowchart.

## 3. Results

*Previous Research on the Sharing Economy*

When sharing platform businesses such as Airbnb and Uber appeared on the scene, the size of the sharing economy began to expand on a full scale, and many analyses and definitions have spread. For example, words such as cooperative, forward market, non-profit market, mass capitalism, open source, on-demand, cooperative consumption, prosumer, online to offline convergence (O2O) platform, platform economy, gig economy, etc. have all been associated with the ideas of the sharing economy as well as the other names of the sharing economy. From collaborative consumption to sharing economy, there is a wide range of definitions with no clear boundaries between them [7]. As Rachel Botsman and Arun Sundararajan have said, there seems to be no shared or agreed definition of sharing economy yet, which means diverse definitions exist [8].

Sharing economy basically starts from the relationship between objects and subjects in economics. It is clearly explained that objects in economics are information, material (resource), and relations (basically time, space, humans), and subjects in economics are supply, demand, and market platform as shown in Figure 3. As a result, diverse definitions of sharing economy differ from objects and subjects. Meanwhile, this could be applied to make an organizing framework that allows mapping out and integrating the different perspectives on the sharing economy. In this Section, this study outlines the criteria for selecting studies for systematic review.

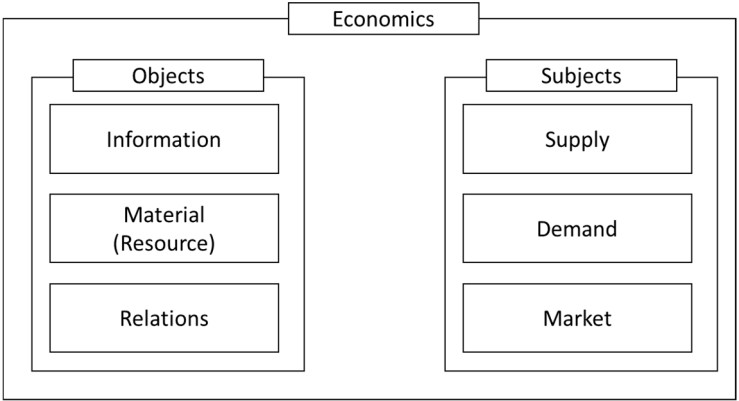

**Figure 3.** Fundamental factors of the sharing economy.

In a historical sense, the sharing economy is not a new concept, since it has existed in the past. During the Middle Ages in Europe, people formed a "guild" to protect their professions or money, which can be considered as an institution of the sharing economy. The first cooperative, which shared consumption and production and operated group buying, was born in Rochdale, Manchester, England.

The words of sharing rides and cooperative consumption appeared for the first time in an article written by Marcus Felson and Joe L. Spaeth [9]. They defined sharing economy as sharing time or objects together in a group of families, friends, etc. Elinor Ostrom explained that "voluntary and autonomous rules are design principles and mechanisms that would allow mutual monitoring and sanctions between members of a community to manage common properties efficiently" [10].

However, the sharing economy in the offline society is limited due to the scarcity of resources; even when a cooperative is engaged in sharing, there is a cost of owning to consider, and the marginal utility of shared values is static. These are perhaps reasons why the concept was indeed limited in practice and ended up playing a niche role in the economy.

Later, Carol Rose suggested the invention of the Internet during the Third Industrial Revolution could lead to a "comedy of commons"—in contrast to the traditional "tragedy of commons"—in which numerous networked computers are brought together like a village festival to create a much greater informational value as a sharing economic network [11]. Online platforms and open sources began to be vitalized as the cost of sharing information reached zero because of the Internet. It was at this time that the idea of "sharing information" online began to be discussed openly and thus greatly expanded the concept of open source. Yochai Benkler predicted that the economic paradigm would be centered around information, culture, education, computing, and communications based on digital platforms like free software, distributed computing, wireless networks, etc. [12]. An information-sharing movement based on copyright called Copyleft was started by Richard Matthew Stallman. This later influenced the rise of the free software movement that emphasized "freedom" when software was being used. Eric Raymond equated the term "free" with "not having to pay" and coined the term "open source". He explained how the process of open source development operated and repeated this in his book The Cathedral and the Bazaar and led the open source movement with this explanation [13]. Henry Chesbrough mentioned that open source software is a collaborative, community model of development [14]. Apache Software Foundation utilized licenses with different levels of strength in terms of regulations, yet still backed the use of open source software and emphasized the need to balance creativity with practice. Lawrence Lessig established Creative Commons, which not only encouraged people to make use of copyrighted materials but also protected authors with six different levels of licenses.

Thomas Eisenmann described the sharing economy platform as a two-sided market where transactions incur costs and benefit from both the sides of buyers and sellers [15]. This analysis of Eisenmann is considered as the most practical approach to platform-based sharing economy experiments such as Uber, Kakao taxi, etc. that came on the scene recently. Alvin Toffler then predicted the coming of "prosumers"—consumers who engage in the production process at the same time to make sure that their needs are reflected [16]; this is exactly one of the most prominent features of the sharing economy that Eisenmann called the "two-sided market" where consumers and producers exhibit characteristics of one another. This trend is called co-creation between customers and companies, which enables open innovation in service [17].

The sharing economy paradigm began to enter the mainstream over the past decade, and the analysis on offline collaborative consumption also began to expand, thus establishing the concept of the sharing economy. In What's Mine is Yours, Rachel Botsman and Roo Rogers divided collaborative consumption into three sections: Redistribution of goods from where they are not needed to where they are needed, collaborative lifestyle, and product service [18]; they declared that the 21st century shall be the age of collaborative consumption. According to Rachel Botsman, collaborative consumption is "based on sharing, selling, renting of goods and services that places access above ownership; it is an economic model that does not just recreate who does the consumption but how it is done".

Meanwhile, in her book The Mesh, Lisa Gansky focused on the "mesh" and claimed that its characteristics are possibility to share, digital network-basis, spontaneity (immediacy—can be shared at anytime, anywhere), and being led by social media platforms [19], and that it is going to be a system that incorporates the whole world. Alex Stephany, the founder of JustPark, who has advanced a definite theory of sharing economy said, "Sharing economy is to find resources that are underutilized and use the accessibility information online to make them available for the community and provide it with opportunities to consume them [20]. The value of the sharing economy is then its ability to reduce the need to own". Steve Schlafman has advocated an on-demand economy where tailor-made services are supplied to consumers utilizing mobile networks, and Uber and Airbnb are also on-demand services that fit the description [21].

According to Daniel Pink, if the sharing economy is seen from the perspective of labor, individuals share idle time, and "calling" starts to replace vocation which leads to the introduction of the concept of gig economy, where freelancers who are professionally committed to vocations to the extent to consider vocations their callings replace professional workers who work for organizations [22]. Tapping the concept of on-demand economy, Denis Johnson and Andrew G. Simpson suggested that transactions in idle labor power are revolutionizing the structure of the labor market [23].

Others have suggested that the sharing economy is a new form of the market or economic phenomenon. In Sharing Economy: An In-depth Look at Its Evolution & Trajectory Across Industries, an article by Michael J. Olson and co-written by Samuel J. Kemp, it is mentioned that the phenomenon of the sharing economy started with individual producers trying to lower costs and raise profits; individuals, enterprises, institutions, etc. that share idle resources and technologies lead to an outcome where both sides, i.e., agents of distribution and agents of usage, receive economic benefits [24]. A Tencent researcher defined sharing economy as an economic phenomenon when one shares idle resources (whoever owns it) with others utilizing community platforms [25]. Arun Sundararajan defined "crowd-based capitalism" as exchanges based on public networks; in short, the true nature of an exchange is considered as an activity that could resolve the contradiction between objective-directed gift economy and the profit-directed market economy. The sharing economy could be approached with Thomas Eisenmann's definition that understands it as the separation of transactions and sharing economies, or with that of Lewis Hyde, which claims the sharing economy is a thought of not only engaging in economic activities but also transmitting social fellowship, mutual reciprocity, and social values.

The following Table 1 summarizes previous definitions searched in chronological order and criteria of objects and subjects.

**Table 1.** Definitions of sharing economy.

|  | **Definitions** | **Objects** | **Subjects** |
|---|---|---|---|
| Marcus Felson, Joe L. Spaeth (1978) | Sharing time or goods in a group, such as family and friends | Time, goods | Demand, supply |
| Elinor Ostrom (1990) | Mutual inspections and sanctions between members of a community are intricately designed institutional measures that allow effective management of commonly owned resources with voluntary and autonomous regulations | Commonly owned resources | Demand, supply |
| Richard Matthew Stallman (1985) | Software source codes are public goods recognized as commonly owned resources; all users are allowed to freely use, analyze, amend, and distribute software | Free software | Demand, supply |
| Yochai Benkler (2005) | Sharing economy expands around digital platforms such as free software, distributed computing, wireless networks, etc. | Free software | Demand, supply |

**Table 1.** *Cont.*

| | Definitions | Objects | Subjects |
|---|---|---|---|
| Rachel Botsman, Roo Rogers (2010) | Redistributing goods from where they are not needed to where they are, cooperative life, collaborative consumption | Goods | Demand, supply |
| Lisa Gansky (2010) | Digital technologies, such as social media platforms, are used as bases to efficiently distribute physical resources | Physical resources | Demand, supply, market |
| Alex Stephany (2015) | Diminishing the need to own resources by finding underutilized resources to provide supply opportunities via online accessibility | Underutilized resources | Demand, supply, market |
| Michael J. Olson, Samuel J. Kemp (2015) | A model stemming from individual producers in an effort to lower costs and maximize profit where distributers and users yield mutual economic benefits by sharing idle resources and technologies among individuals, enterprises, institutions, etc. | Idle resources | Demand, supply |
| Tencent (2016) | Sharing economy is an economic phenomenon where the public create income by sharing idle resources, no matter who owns them, by taking advantages of community platforms | Idle resources | Demand, supply, market |
| Baojun Jiang, Lin Tian (2016, 2018) | A mechanism to determine the sharing price in sharing economy. Sharing idle resources that result from consumer-to-consumer relationships | Idle resources | Demand, supply, market |
| Danial Pink (2001) Denis Johnson, Andrew G. Simpson (2015), Guda, Subramanian (2017) | Geek economy and transactions in idle labor power. An on-demand platform-based work environment where participation and work are voluntary instead of set working hours or contract | Geek economy, idle labor power | Demand, supply, market |
| Arun Sundararajan (2016) | Crowd-based capitalism, where the economic agent becomes the public instead of enterprises; this is a paradigm where economic and social elements are mixed | Commercial exchange, labor | Demand, supply, market |

To sum up, sharing economy has become catch-all label both as an umbrella construct and a contested concept [26]. Each scholar has each perspective regarding the sharing economy despite the similar concepts for sharing activities themselves. There are difficulties in understanding sharing economy as an existing economy because there are limitations in defining diverse and multidimensional characteristics of sharing economy comprehensively. The previous researchers' definitions also lack consistency.

In order to include diverse concepts that were presented previously, Martin Weitzman reflected the multidimensional nature of the sharing economy as he classified sharing in terms of objectives and targets [27]. According to Weitzman's definition, businesses that engage in the sharing economy are classified into asset rentals and service providers and distributed as sales or exchange depending on their methods of sharing. However, many sharing platform startups that do not easily conform to this classification scheme are emerging, and the objects of consumption or sharing and market are treated separately, so the deduction of an overarching definition and policy is not easily attained.

Therefore, Sofia Ranchordas argued that a mistake such as insinuating innovative services into the existing legal structural framework should not be made, and regulations regarding the sharing economy should be approached with the perspective of legal innovations [28]. Now an all-inclusive model that could integrate all the existing diverse concepts of sharing economy is required. This would assist the fundamental comprehension of the sharing economy and would serve as a foundation when establishing national strategies for the sharing platform economy.

## 4. Discussion

### 4.1. Definition of Sharing Economy

This study defines sharing economy as "a series of activities that leads to value creations as economic agents share economic objects and delineate sharing objects of information, materials, and relationships as sharing between economic agents in the markets of supply and demand. At this time, this model starts based on a platform for sharing just like a platform business model [29]. In short, the objects of a sharing economy (information, materials, relationships) and the agents of a sharing economy (supply, market platform and demand) are matched via three market platform (online platform, O2O platform, gig platform), thus forming a 3 × 3 matrix. This model is called the Sharing Economy Cube Model (2018) and is classified as profit or non-profit depending on the aim and pursuit of profits. In the end, the discourse on sharing economy will be about: (1) what will be shared? (Objects); (2) who will own them? (Agents); (3) why it is shared? (Objectives). There could be many different permutations depending how these questions are answered, but through the 3 × 3 × 2 integration model, as Figure 4 shows.

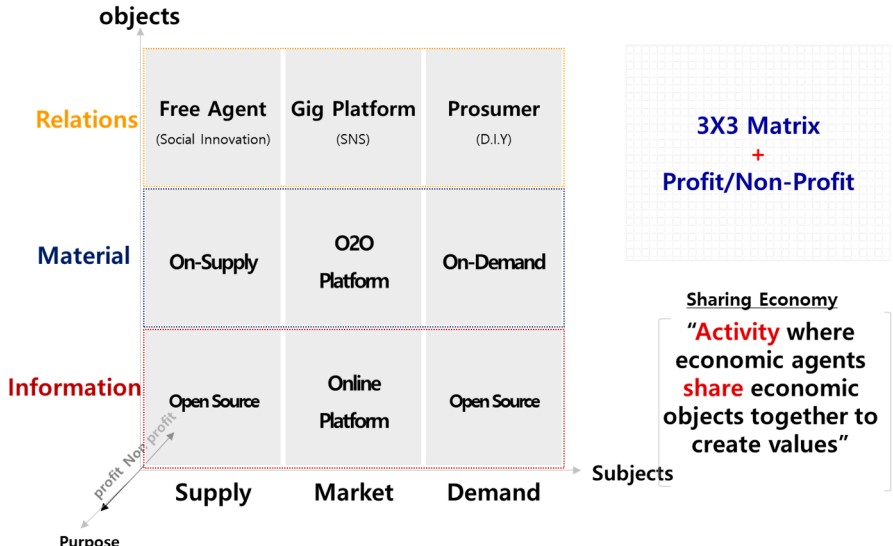

**Figure 4.** KCERN Sharing Economy Cube Model.

While previous studies draw a wide range of definitions of sharing objects, such as open source, spaces, resources, and relations in the side of supply and demand, this model improves the suggestion of market as the platform for network and sharing in the middle. Moreover, objectives of profit or non-profit are added factors that previous studies did not consider. Of course, two purposes might be combined or involved in cyclical dynamics in open innovation [30]. This is a big difference from the advanced definitions and network economy.

### 4.2. Implications and Limitations

There are three implications of this unified model; explaining relationships between industrial revolution and sharing economy, unification of previous definitions, and drawing national strategies to develop the sharing economy. However, this model still has limitations of generalization.

#### 4.2.1. Relationship between Industrial Revolution and the Sharing Economy

KCERN cube model can explain the evolution of industrial revolution and the sharing economy, which previous definitions are incapable of doing, and accurately reflects the true nature of the sharing economy, which is sharing that occurs between economic agents and objects. The significance of this model is that the evolutionary stage of industrial revolution and the relationships in the

sharing economy are reinterpreted as the evolutions of platform and economic objects. The First and Second Industrial Revolutions were material (physical) revolutions offline and the ownership economy consisted most of the economy, which meant that the sharing economy comprised an insignificant amount of the economy. After then, the Third Industrial Revolution created internet-based online platforms as the wired internet developed, which enabled sharing of information to be active [31], which is connected to first layer of KCERN cube model. The internet revolution lowered the cost of sharing to the extent that the marginal cost became zero [32] and sharing activities of information began to increase exponentially and the law of increasing marginal utility took hold.

During the Fourth Industrial Revolution combining the online worlds and offline worlds (i.e., O2O integration), reality and the virtual world began to merge as well; offline economy started to be sharing-economized due to the O2O fusion. Later, there were opinions that suggested the progress of a blockchain technology-based sharing economy that was segregated from platform operators would also be part of this evolution [33].

### 4.2.2. Unification of Previous Definitions

Secondly, the core of this model exceeds the limitations in preexisting concepts and definitions; it enables the fundamental and integrative analysis of the sharing economy by understanding the expansion of the sharing economy. As shown in Figure 5, all the diverse and pre-existing definitions of sharing economy are included in the KCERN Sharing Economy Cube Model. If any new form of sharing economy arises, it would be categorized and explained using KCERN sharing Economy Cube Model.

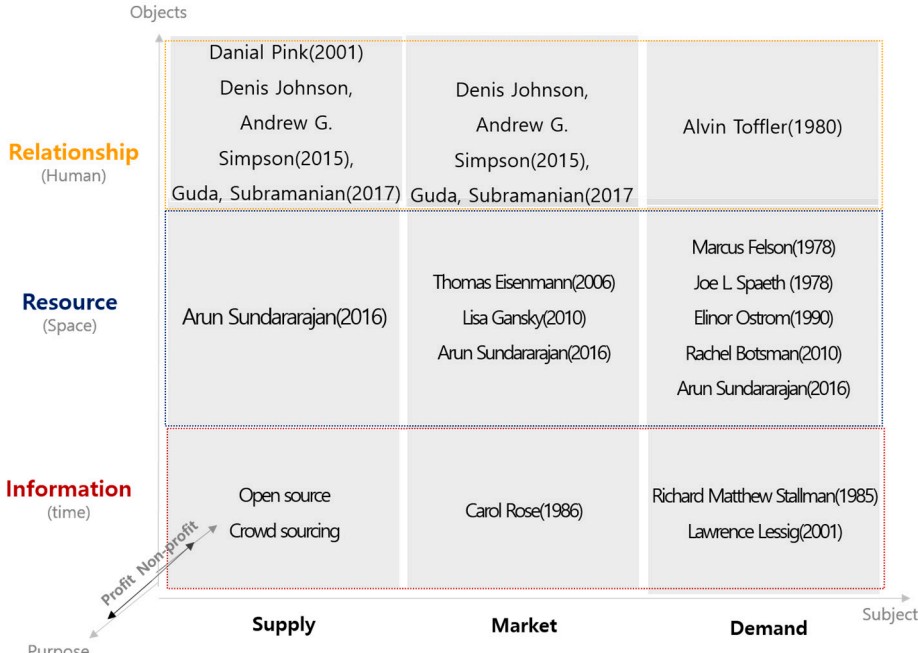

**Figure 5.** Previous definitions in KCERN Sharing Economy Cube Model.

Recently growing sharing platform companies or businesses also can be applied based on each component of the KCERN model as Figure 6 shows. For example, development or utilization of open sources occurring on online platforms such as Github and Apache would be examples of information sharing. Next, sharing of materials on O2O platforms, Wework, Airbnb, etc., would be representative examples that we are already familiar with. Lastly, sharing of relations on gig platforms such as TaskRabbit, Upwork, etc. are leading the gig economy.

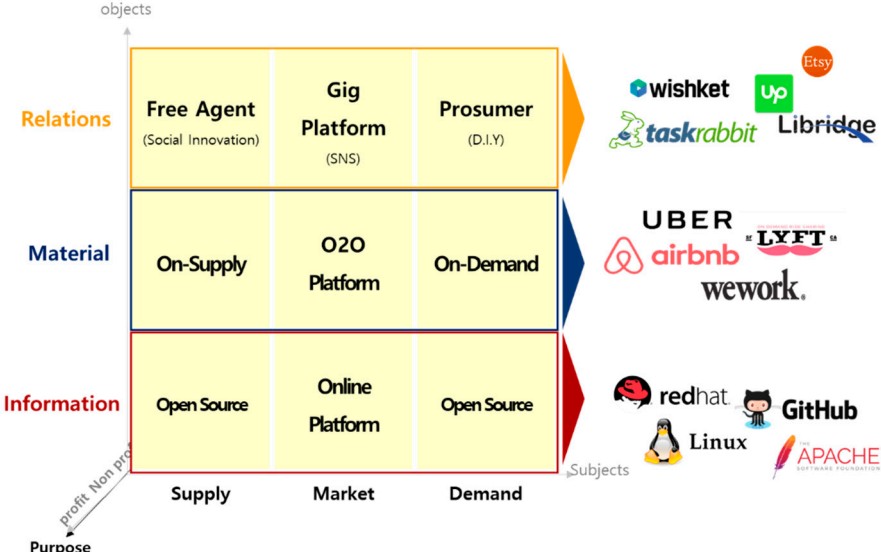

**Figure 6.** Example based on KCERN sharing economy model.

It is hoped and expected that not a specific model but a unified model will contribute to form an integrative consensus of the sharing economy fast becoming a mainstream economy. However, it is required for universal validity to review whether this model can accept all the existing or newly appearing concepts.

### 4.3. National Sharing Economy Strategies

There were difficulties in making national strategies consistent because many neighboring concepts on the sharing economy were disordered. However, this suggested integrative sharing economy model could draw an overall national strategy based on sharing factors of information, material, and relations, the fundamental factors of economics in stages. It would not be limited to a nation, but it is necessary to be reflected upon when building a roadmap for national strategy.

National strategies can be deducted to vitalize the sharing economy based on an integrative model of sharing economy. However, individual or phenomenal approach without the comprehension of the nature of the sharing economy has limitations. There is a need for an integrative perspective that is based on the insight that the sharing economy starts with sharing of information and proceeds to the sharing of materials and relationships. In this study, we propose to build a policy roadmap and three strategies that expands sharing of information, materials, and relationships based on the KCERN Sharing Economy Cube Model.

First, information exchange is the starting point of the expansion of sharing economy and the catalyst for innovations. Secure utilization of information should flourish as data become open and shared.

Therefore, regulative reforms are needed in order to vitalize sharing of information and to build the open source ecosystem. The U.K. first classified data into three level of data standards, which are (1) official (2) secret and (3) top secret. In order to build an open source ecosystem, a market where developers can freely participate in tens of millions of open source projects should be constructed. It is also necessary to vitalize communities and spaces, like GitHub, where developers can communicate and share their ideas.

Restrictions on utilizing de-identified personal data anonymity can allow coexistence of proper utilization and security of private data if control over private information is strengthened, but principles such as explicit agreement and strict prior notification are softened while turning to a system that holds a violator responsible, with severe penalties, for reidentification.

Second, as the Fourth Industrial Revolution has emerged, the sharing of materials on O2O platforms has expanded, so utilization of cloud computing, which enables such expansion, should be facilitated to organize diverse foundations for convergence. Regulatory reforms in the cloud industry need to be implemented to lower barriers of entry to the cloud industry, and policies deigned to vitalize the cloud market need to be prepared.

Since the systems that protect existing offline business operators could impede and inhibit expansion of sharing economy, the first priority of national policies should be welfare of consumers.

There are three recommendations as remedies for excessive concentration of power among sharing platform companies, whose share of the world economy reaches about 50%: (1) heavy separate taxation on earnings from non-innovative activities; (2) multihoming policies that maintain competition in the platform industry; (3) policies to ensure transparency in operation using the blockchain technology.

Third, knotworking and gig economy are creating many new jobs by alleviating the imbalance between supply and demand for specific technologies or abilities by making use of gig platforms. The U.S. Department of Labor considers a shared economy worker as an independent contractor [34] or a sole proprietor and interprets collateral side payments in the two-sided market through various platforms. During the process of expanding labor flexibility, absence of an effective insurance system, which serves as a safety net, could discourage the will to retry. The whole labor market should be flexible, but there should be infrastructures, safety nets, matching platforms, and retraining programs in place to ensure that individuals' jobs can securely be relocated. Table 2 shows three strategies for a national roadmap.

**Table 2.** Major national strategies to vitalize the sharing economy.

| Sharing of Information | Sharing of Material | Sharing of Relations |
| --- | --- | --- |
| Classify and open public data Eliminate excessive security-first policies Supply incentives to open sourcing efforts Build open source ecosystems Balance protecting and utilizing private information Regulate reidentification Give the ability to control one's private information | Eliminate offline entry regulations Turn towards negative regulative regime Heavy separation tax on rental earning Maintain competitiveness in multihoming Operation transparency based on blockchain technology | Redefine "work" Vitalization of gig platforms Prepare job safety nets and reeducation system |

This is expected to serve as basic guidelines for most countries which build national strategies while placing different priorities on factors corresponding to each country's condition.

## 5. Conclusions

This study was an attempt to offer national strategies to plan a path toward the sharing economy with the KCERN Sharing Economy Cube Model. Existing definitions and analyses of sharing economy lack consistency and integrity with their approaches based on specific phenomena, thus possessing limitations in deriving policies. However, this study offered a unified model, reflected diverse and complex existing concepts, and reinterpreted the sharing economy in relation to the evolutionary trajectory of industrial revolution. Furthermore, the model offers a national strategic roadmap to as a basis for policies.

The growth of the sharing economy is depicted through three stages: (1) internet-based information sharing expanded through digital transform during the Third Industrial Revolution; (2) the development of the sharing economy centers around the expansion of sharing of materials and relationships, particularly due to analog transformations in Fourth Industrial Revolution; (3) ultimately,

the blockchain-based technologies of trust leads to the convergence the for-profit and not-for-profit sharing economies.

This study presents the three major national strategies based on the KCERN model as follows: (1) remove excessive security-first policies and build an open source ecosystem that starts from sharing of information; (2) remove offline entry barriers and initiate regulatory reforms to stimulate cloud computing to expand sharing of materials; (3) prepare retraining systems and job safety nets to ease sharing of relationships. In short, these are the three vital areas of the sharing economy.

The main purpose of this study is to present the integrative model and concept of the sharing economy, which are used as bases to gain fundamental understanding of the sharing economy, but it is not quite sufficient to resolve the controversies and conflicts that exist around the platform-based sharing economy. Therefore, there is a need for further research on the desirable evolutionary direction of the sharing platform economy.

**Author Contributions:** Conceptualization and methodology, M.L.; formal analysis, investigation, writing—Original draft preparation, Y.K., M.L.; writing the paper, Y.K.; supervision and project administration, M.L. All authors read and approved the final manuscript.

**Funding:** This research received no external funding.

**Acknowledgments:** This study basically draws from the authors' forum report. All remaining errors are the responsibility of the authors.

**Conflicts of Interest:** The authors declare no conflict of interest.

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
