# Peer review of "Typology and Unified Model of the Sharing Economy in Open Innovation Dynamics"

_2199-8531, doi:10.3390/joitmc5040102_

Round 1

Reviewer 1 Report

Since this study is the result of the Typology of sharing economy, it needs to be drastically modified to match the results of the research carried out.

1. It is recommended that the title be changed to 'Typology of sharing economics'.

2. It is recommended that Table 1 be deleted as it is not very helpful to understand.

3. The contributions of this study are the 'Figure 2. Findings' and 'Figure 2. KCERN Sharing Economy Cube model'.

   A full revision is needed to ensure that these contributions are well understood.

4. It is recommended that the structure of the article be modified as follows:
   Induction/Methods/Results/Discussion/Conclusion

5. Prove the logic that the literature of Table 2 in addition to the study qustions is suitable for the study workflow.

   It is necessary to work out specific procedures for Systematic Reviews.

   PRISMA statement is typically used among Systematic Reviews.

   Please refer to the following literature.

    PLoS Med 6(7):e1000097. https://doi.org/10.1371/journal.pmed.1000097.

   Research questions currently presented in this process will also have to be revised.

6. Results are recommended to be revised in full, focusing on the following:

   6-1. Demonstrate 'Figure 2. Findings'. In this process, the literatures in Table 2 should be actively utilized.

   6-2. Demonstrate 'Figure 2. the KCERN Sharing Economy Tube model' in detail. In this process, the literatures in Table 2 should be actively utilized.

   6-3 After combining Figure 3 and Figure 5, the practical advantages of the KCERN Sharing Economy Cube Model should be justified by examples.

7. Re-configure Discussion as implication and limitation.

    7-1. implication

         3.3. Briefly describe the details of '3.3. Relationship Between Industrial Revolution and sharing economics' and '4. National Sharing Economy Strategy'.

               There is a bit of exaggerated interpretation in the first draft, so make sure to make a big reduction.

               Figure 4 appears to be a pile, so I recommend deleting it.

    7-2 Limitation

        It is recommended that you rewrite the limitations of the study.

        Please state that the implications of the Industrial Revolution and the National Sharing Economy Strategy are subjective interpretations by the authors.

8. Please correct various errors that degrade readability.

   An awkward part is found in Caption Number, Grammar, etc.

   In particular, it is difficult to follow references in two ways. It is proposed to unify to either (the author, the year) or [number].

Author Response

It is recommended that the title be changed to 'Typology of sharing economics'. -

"Typology of Sharing Economy and Unified Model"

2. It is recommended that Table 1 be deleted as it is not very helpful to understand. - Delete

3. The contributions of this study are the 'Figure 2. Findings' and 'Figure 2. KCERN Sharing Economy Cube model'.
   A full revision is needed to ensure that these contributions are well understood.
  -Done. Please see the revised version. 

4. It is recommended that the structure of the article be modified as follows:
   Induction/Methods/Results/Discussion/Conclusion 

  -Done. Please see the revised version. 

5. Prove the logic that the literature of Table 2 in addition to the study qustions is suitable for the study workflow.

   It is necessary to work out specific procedures for Systematic Reviews.

   PRISMA statement is typically used among Systematic Reviews.

   Please refer to the following literature.

    PLoS Med 6(7):e1000097. https://doi.org/10.1371/journal.pmed.1000097.

   Research questions currently presented in this process will also have to be revised.

-  -Done. Please see the revised version. 

6. Results are recommended to be revised in full, focusing on the following:

   6-1. Demonstrate 'Figure 2. Findings'. In this process, the literatures in Table 2 should be actively utilized.  -Done. Please see the revised version. 

   6-2. Demonstrate 'Figure 2. the KCERN Sharing Economy Tube model' in detail. In this process, the literatures in Table 2 should be actively utilized.  -Done. Please see the revised version. 

   6-3 After combining Figure 3 and Figure 5, the practical advantages of the KCERN Sharing Economy Cube Model should be justified by examples.  -Done. Please see the revised version. 

7. Re-configure Discussion as implication and limitation.

    7-1. implication

         3.3. Briefly describe the details of '3.3. Relationship Between Industrial Revolution and sharing economics' and '4. National Sharing Economy Strategy'.   -Done. Please see the revised version. 

               There is a bit of exaggerated interpretation in the first draft, so make sure to make a big reduction.  -Done. Please see the revised version. 

               Figure 4 appears to be a pile, so I recommend deleting it.  -Done. Please see the revised version. 

    7-2 Limitation

        It is recommended that you rewrite the limitations of the study.

        Please state that the implications of the Industrial Revolution and the National Sharing Economy Strategy are subjective interpretations by the authors.  -Done. Please see the revised version. 

8. Please correct various errors that degrade readability.  -Done. Please see the revised version. 

   An awkward part is found in Caption Number, Grammar, etc.

   In particular, it is difficult to follow references in two ways. It is proposed to unify to either (the author, the year) or [number].

Reviewer 2 Report

It deservers the publication.

Author Response

Revised version uploaded 

Round 2

Reviewer 1 Report

This study provides an analytical tool for studying the sharing economy by classifying the types of sharing economy.
I believe this new trial will be an important reference for the researchers involved.

Author Response

Revised for systematic thesis structure. 

This manuscript is a resubmission of an earlier submission. The following is a list of the peer review reports and author responses from that submission.

Round 1

Reviewer 1 Report

In introduction, the authors mentioned the 4th industrial revolution without any academic reference. It has received attention from public since introduced at WEF. However, in an academic research paper, the manuscript needs more academic background of the term to use it. The authors should find more references on the industrial revolution including 1st to 4th. The authors summarized the milestones of sharing economy very well. If the authors have some difficulties in finding concrete academic background of the industrial revolution, it is a better way to focus on sharing economy and the model besides the 4th industrial revolution in the manuscript.

The author must check the reference very seriously. For instance, the reviewer cannot find WFE 2017 in the reference section even though it is cited in the introduction part. PWC 2014 might be Hawksworth et al. (2014) of the reference part, is it correct? There are two KCERN (2018) in the reference list. The readers cannot distinguish them unless it is designated like 2018a and 2018b. Please review the current version of manuscript if it meets the formatting criteria of the journal.

The authors offer the KCERN Sharing Economy Cube Model, as mentioned in the manuscript. However, the reviewer is wondering if the model is offered in the current paper first. All the figures (Figs. 1-3) cite KCERN (2018). The model might be introduced at a forum or conference already? If so, it should be mentioned in the manuscript. In this case, the model was proposed at the event, and the authors should write the current paper with more concrete academic background and/or investigation. It will support the KCERN model strongly. Anyway, the history of the model should be clarified in the manuscript.

Line 166: KCERN (2016) is a typo, or the model was originally proposed in 2016 by KCERN?

Author Response

Please review this paper, which reflect your comments somehow.

Reviewer 2 Report

This study introduces the Sharing Economy Cube Model.

Recently, various attempts have been made to systematically understand the Sharing Economy, and this research is on its way.

I agree with the motivation to research that the concept of Sharing Economy needs to be formulated systematically, but I have some deep concerns about this paper.

1) The authors do not clearly define the controversial concept, but develop the writing.
   Examples of terms used differently for different users include sharing economy, 3rd industrial revolution, 4th industrial revolution, and national strategy.

2) Literature review of existing academic researches on sharing economy in similar motive of authors was not conducted.
   As I know, the existing literature that explores sharing economy in similar motives to authors is as follows.
   Technological Forecasting & Social Change 125 (2017) 11-20.
   Environmental Innovation and Societal Transition 23 (2017) 70-83.
   Environmental Innovation and Societal Transition 23 (2017) 92-104.
   Journal of Cleaner Production 212 (2019) 1154-1165.

3) As a result, knowledge gap and implication are abstract.
   What difficulties do you experience when understanding sharing economy as an existing economic or business concept?
   How have existing academic literature on the concept of sharing economy defined the research questions and made progress?
   Still, what research questions have not yet been resolved?
   How does the Sharing Economy Cube Model overcome the limits of existing research?
   What are the limitations of the Sharing Economy Cube Model?
   How can the Sharing Economy Cube Model be used in other subsequent studies, and what is its academic contribution?

4) The authors confuse the concept of sharing economy and network economy in many places.
   There are similarities between the two concepts, but they are different.
   At least the Sharing Economy Cube Model offers a lot of scope to the concept of network economy rather than sharing econmoy.

5) Methodology was not systematically presented. In addition, the process of inducing the Sharing Economy Cube Model and its proof were not presented.

6) There is a logical gap between Sharing Economy and National Strategy.

   It can go with the political slogan of any country. However, setting the boundary of 'nation' in the sharing econmoy is not consistent with the concept of sharing.

   How can two concepts be naturally linked?

Author Response

(The authors gave the same response as above.)

Round 2

Reviewer 1 Report

The reviewer still does not agree that the manuscript meets the criteria for the publication. Please read the instruction for the authors very carefully. ( https://www.mdpi.com/journal/JOItmC/instructions ) The format of reference is incorrect. Some references were written by the same author, Minhwa. In the current manuscript, three versions (Minhwa, Min Hwa, MinHwa) are used. Besides this, there are many errors. 

What is line 162? 

Please remind that this is not an assignment in classroom, but an academic paper in a highly standard journal. 

Reviewer 2 Report

The authors performed a revision that was beyond the intent of the comments I pointed out.
The problem of this manuscript is as follows.
1) The composition of the text is not logical and is full of the authors' arguments.
2) The knowledge gap of previous studies is not explained in detail.
3) There is no logical proof of how this study solves the knowledge gap of previous studies.
4) The authors are arguing without a presumption of universal consensus on controversial concepts.
   In this manuscript, the most controversial concepts are industrail revolution, sharing econmoy, national strategy.
5) The adoption of the sharing economy as a national strategy is a local topic of a particular country, but it is an inappropriate topic for international academic journals seeking universal values.
 - The adoption of the sharing economy as a national strategy is a matter that depends on the economic, social and cultural context of the country. It is a matter of value judgment that is difficult to have universality from an international point of view. This can be a political slogan of a particular country, but it is a disgraceful subject for international academic journals seeking universal values. At this stage, conceptual discussions should be made faithfully enough to fit at least the international academic journals. Yet, discussions of concepts and advances in knowledge have not matured enough to be published in academic journals.